# Evaluating Leukocyte Telomere Length and Myeloid-Derived Suppressor Cells as Biomarkers for Prostate Cancer

**DOI:** 10.3390/cancers16071386

**Published:** 2024-03-31

**Authors:** Haruhiko Wakita, Yan Lu, Xiaoxu Li, Takuro Kobayashi, Tsuyoshi Hachiya, Hisamitsu Ide, Shigeo Horie

**Affiliations:** 1Department of Urology, Graduate School of Medicine, Juntendo University, Tokyo 113-8431, Japan; h-wakita@juntendo.ac.jp (H.W.); lyan@juntendo.ac.jp (Y.L.); x.li.ko@juntendo.ac.jp (X.L.); ta-kobayashi@juntendo.ac.jp (T.K.); h.ide.me@juntendo.ac.jp (H.I.); 2Department of Advanced Informatics for Genetic Disease, Graduate School of Medicine, Juntendo University, Tokyo 113-8431, Japan; t.hachiya.wa@juntendo.ac.jp; 3Department of Digital Therapeutics, Graduate School of Medicine, Juntendo University, Tokyo 113-8431, Japan

**Keywords:** prostate cancer, leukocyte telomere length (LTL), myeloid-derived suppressor cell (MDSC), aging

## Abstract

**Simple Summary:**

Prostate cancer is the most common cancer in men. The pursuit of novel biomarkers for early detection of prostate cancer poses a contemporary challenge given the age-associated escalation in prostate cancer risk and severity. Our focus was directed towards evaluating leukocyte telomere length (LTL) and myeloid-derived suppressor cells (MDSC) in prostate cancer patients, considering their potential as adjunctive diagnostic markers. In a study of 102 patients who underwent prostate biopsy, those diagnosed with prostate cancer demonstrated significantly shorter LTL and an increased proportion of M-MDSC prior to diagnosis, along with elevated PSA levels and age, in comparison to controls. Furthermore, a significant negative correlation was observed between LTL and MDSC levels. This initial report of those findings could potentially contribute to a deeper understanding of the molecular, biological, and immunological factors involved in cancer development.

**Abstract:**

Background: Leukocyte telomere length (LTL) and myeloid-derived suppressor cells (MDSC) are associated with aging and the development and progression of cancer. However, the exact nature of this relationship remains unclear. Our study aimed to investigate the potential of LTL and MDSC as diagnostic biomarkers for prostate cancer while also seeking to deepen our understanding of the relationship of these potential biomarkers to each other. Methods: Our study involved patients undergoing a prostate biopsy. We analyzed the relative LTL in genomic DNA obtained from peripheral blood leukocytes as well as the percentage of MDSC and their subtypes in peripheral blood mononuclear cells (PBMC). Our evaluation focused on examining the relationship between LTL and MDSC and pathological diagnoses as well as investigating the correlation between LTL and MDSC levels. Results: In our study of 102 participants, 56 were pathologically diagnosed with localized prostate cancer (cancer group), while 46 tested negative (control group). The cancer group exhibited significantly shorter LTL in comparison to the control group (*p* = 0.024). Additionally, the cancer group showed a tendency towards a higher percentage of monocytic MDSC (M-MDSC), although this difference did not reach statistical significance (*p* = 0.056). Our multivariate logistic regression analysis revealed that patients with shorter LTL and higher percentages of M-MDSC had a 2.98-fold (95% CI = 1.001–8.869, *p* = 0.049) and 3.03-fold (95% CI = 1.152–7.977, *p* = 0.025) increased risk of prostate cancer diagnosis, respectively. There was also a significant negative correlation between LTL and M-MDSC. (r = −0.347, *p* < 0.001). Conclusions: Our research has established a correlation between LTL and MDSC in patients undergoing biopsy for prostate cancer. Notably, we observed that individuals with localized prostate cancer tend to have shorter LTL and a higher percentage of M-MDSC prior to their diagnosis. These findings suggest that LTL and M-MDSC could potentially serve as adjunctive biomarkers for the early diagnosis of prostate cancer.

## 1. Introduction

Telomeres are repetitive DNA sequences consisting of TTAGGG located at the ends of each chromosome. They are involved in protecting chromosomes from degradation and instability [1,2]. Telomeres, which shorten during cell division or proliferation, play a crucial role in cellular aging and stability. When telomeres are reduced to a certain length, cells generally cease dividing and enter a state of senescence or undergo apoptosis. However, in certain instances, the chromosomal instability (CIN) resulting from this telomere shortening can contribute to cancer development [3,4,5]. There has been a suggestion that CIN closely interacts with chronic inflammatory and immune responses in the tumor microenvironment (TME), which promotes tumor growth [6]. 

In this study, we examined myeloid-derived suppressor cells (MDSCs), a key component of the tumor microenvironment (TME). MDSCs are pivotal in regulating immune responses in cancer pathogenesis, infections, chronic inflammation, and traumatic stress. They contribute to cancer progression by suppressing effector T cells, crucial for the antitumor immune response [7,8,9]. Additionally, MDSCs serve as markers of immunosenescence, with their levels increasing with age [10,11,12]. There has been a dramatic increase in reports providing epidemiological, experimental, and clinical evidence of the association between telomeres and MDSCs of each other and cancer.

Cancer is the disease of most significant concern in developed countries. While research is being conducted around the world, and cancer treatment is making remarkable progress, one of the societal problems is the economic burden placed on the world by the increasing number of new cancer patients and the rising cost of cancer treatment [13]. In particular, the incidence of prostate cancer has rapidly increased due to an increasing elderly population, the prevalence of prostate-specific antigen (PSA)-based screening tests, and advances in diagnostic technology. Prostate cancer is one of the most common cancers among men, has the highest incidence in 112 countries, and is the leading cause of cancer-related deaths in 48 countries, according to 2020 statistics [14]. In Europe, in 2009, the cost of cancer, including medical costs for prostate cancer, accounted for 7% of all cancers after lung, breast, and colorectal cancer [13]. In prostate cancer, short telomere length is a risk factor for incidence and mortality [15,16,17], and a higher percentage of monocytic MDSC (M-MDSC), a subtype of MDSCs, is associated with poor prognosis [18,19,20,21]. However, to our knowledge, a direct correlation between telomeres and MDSCs has yet to be examined in vivo or in vitro.

In this study, we concurrently measured LTL and the percentage of MDSCs in the peripheral blood of patients undergoing prostate biopsy. We also performed the first analysis, to the best of our knowledge, of a direct correlation between telomere shortening and MDSCs, providing clinical validation for our findings.

## 2. Materials and Methods

### 2.1. Study Sample and Data Collection

Our single-center prospective study was conducted at Juntendo University Hospital from 2019 to 2021. A total of 119 patients admitted for prostate biopsy were enrolled in this study. All subjects were suspected of prostate cancer based on one or more of the following: a high PSA level (more than 4 ng/mL), an abnormal digital rectal examination, and/or a Prostate Imaging-Reporting and Data System (PI-RADS) category of 3 or higher for prostate MRI. After excluding 17 patients with distant metastases or double cancers, a total of 56 patients with localized prostate cancer (cancer group) and 46 patients with no cancer (control group) detected on biopsy were included in the study. Blood samples were collected from each patient the day before the biopsy, and LTL and the percentage of MDSCs were measured and analyzed. All participant characteristics, including age, BMI, smoking history, past medical history, and medications, were recorded at enrollment.

### 2.2. LTL Measurement

Relative LTL was measured from genomic DNA with quantitative PCR (qPCR) assay. Genomic DNA was extracted from peripheral blood leukocytes using a Genomic DNA purification kit (Promega, Madison, WI, USA). All qPCR assays were carried out in 96-well plates, using the QuantStudio^®^ 3 real-time PCR system (Thermo Fisher Scientific, Waltham, MA, USA), following the procedures of previous reports with minor modifications [22,23]. The PCR was performed as follows: 15 min at 95 °C, followed by 2 cycles of 15 s at 95 °C and 15 s at 49 °C, and 35 cycles of 15 s at 95 °C and 30 s at 60 °C. Each 20 μL reaction volume contained a 15 μL telomere or single-copy gene (human β-globin, HBG) master mix with 5 μL of DNA at a concentration of 2 ng/μL. A 5-point standard curve was generated by serially diluting the reference genomic DNA (Human Genomic DNA; G152A, Promega, Fitchburg, WI, USA) from 10.25 ng/μL to 0.64 ng/μL. The ratio of the telomeric repeat copy number (T) to the single-copy gene number (S) using the standard curve was represented as the T/S ratio. Relative LTL was obtained by dividing the T/S ratio of each sample DNA by the T/S ratio of the reference DNA. Each sample was measured using three wells on one plate and run twice on a different plate, with the average value calculated as the LTL to ensure improved accuracy and reproducibility of LTL values. Statistical analysis of accuracy showed that the mean intra-assay CV, inter-assay CV, and intra-class correlation (ICC) were 2.5%, 3.7%, and 0.93, respectively.

### 2.3. MDSC Measurement

MDSCs can be categorized into two main subsets based on phenotypic and morphologic characteristics of myeloid markers: polymorphonuclear-MDSC (PMN-MDSC) and monocytic-MDSC (M-MDSC) [24,25,26]. MDSCs were detected from fresh peripheral blood mononuclear cells (PBMC) isolated from peripheral blood by density gradient centrifugation using Histopaque^®^-1077 (Sigma-Aldrich, St. Louis, MO, USA). First, 1 × 10^6^ single cells were suspended in 100 μL PBS and incubated with FcR blocking reagent (Biolegend, San Diego, CA, USA) for 15 min at room temperature, followed by a proper concentration of fluorescent-conjugated antibody in 100 μL PBS for 15 min at 4 °C. The fluorochrome-labeled antibodies for detecting cell surface antigens were CD14-PerCP-Cy5.5, CD15-APC-Cy7, CD33-PE-Cy7, and HLA-DR-PE-Texas Red (Biolegend, San Diego, CA, USA). The labeled cells were washed twice and resuspended in 500 μL buffer with DAPI (1 μg/mL). FACS data were acquired using a BD^®^ LSR II Flow Cytometer (BD Biosciences, San Jose, CA, USA) with BD FACSDiva™ Software and analyzed using Flowjo Software (Tree Star Incs, Ashland, OR, USA). Total MDSC, PMN-MDSC, M-MDSC, and eMDSC (early-stage MDSC) were characterized as HLA-DR^low/−^ CD33^+^, HLA-DR^low/−^ CD33^+^ CD15^+^ CD14^−^, HLA-DR^low/−^ CD33^+^ CD15^−^ CD14^+^, and HLA-DR^low/−^ CD33^+^ CD15^−^ CD14^−^ as a percentage of total PBMC, respectively. The gating strategy for MDSC is provided in Appendix A. Great care was taken to avoid inactivation due to freeze–thaw process or time lapse, and assays of MDSC were performed using fresh samples immediately on the day of sample collection.

### 2.4. Statistical Analysis

Continuous variables were expressed as mean ± SD, Student’s *t*-test and one-way analysis of variance were used to analyze differences between two or three groups for normally distributed continuous variables, and the Mann–Whitney U test and Kruskal–Wallis test were used to compare non-normally distributed continuous variables. The distribution of variables was evaluated by the Shapiro–Wilk test. Pearson’s correlation coefficient or Spearman’s rank correlation coefficient according to normality and partial correlation analysis was used to evaluate correlations between variables. The χ-square test was used to evaluate differences in categorical variables between subgroups. Multivariate logistic regression analysis was used to calculate and evaluate odds ratios and corresponding 95% confidence intervals for the relationship between prostate cancer diagnosis and LTL and MDSC. For confounding factors, we selected age [27], BMI [28], and smoking [29], as reported in previous LTL studies, plus PSA levels most associated with prostate cancer. All statistical analyses were conducted with IBM SPSS (version 28.0), with * a *p*-value ≤ 0.05 (two-tailed) considered to indicate statistical significance.

### 2.5. Ethics Statement

The study was approved by the Institutional Review Board of Juntendo University Hospital (approval number: M19-0158, M20-187), and all experiments were performed in accordance with approved guidelines. All participants provided written informed consent.

## 3. Results

### 3.1. Patients’ Characteristics

All participants were Japanese. Epidemiological and clinical characteristics are summarized in Table 1 and Appendix A. The cancer group exhibited a notably higher age (*p* = 0.005), elevated PSA levels (*p* = 0.027), and increased CRP levels (*p* = 0.050) compared to the control group.

### 3.2. Prostate Cancer Patients Have Shorter Leukocyte Telomere Lengths

The overall mean relative LTL assessed in this study was 0.980 ± 0.142. LTL was significantly correlated with HbA1c level (ρ = −0.292, *p* = 0.006) and renal function (cystatin C (ρ = −0.202, *p* = 0.042)) but not with PSA level (Appendix A). LTL in the cancer group was significantly shorter than in the control (*p* = 0.024). In the cancer group, LTL was associated with the Gleason score and D’Amico risk classification. Patients with a Gleason score of 7 exhibited significantly shorter LTL than those with a score of 6 (*p* = 0.034). While patients with a Gleason score of greater than 8 also showed a shorter LTL than those with a score of 6, the difference was not statistically significant (*p* = 0.132). Regarding the D’Amico risk classification, LTL was significantly shorter in the intermediate (*p* = 0.022) and high-risk groups (*p* = 0.028) compared to the low-risk group (Appendix A). The continuous variable of LTL was categorized into tertiles: short group (<0.900, n = 34), middle group (0.900–1.035, n = 34), and long group (>1.035, n = 34). (There were no significant differences in age or PSA levels between these groups.) Multivariate logistic regression analysis showed that the short group had a significantly higher risk of prostate cancer than the long group (OR = 2.979, 95% CI: 1.001–8.869, *p* = 0.049) (Table 2).

### 3.3. Prostate Cancer Patients Have a Higher Percentage of M-MDSC

The overall mean percentages of total MDSC, PMN-MDSC, and M-MDSC evaluated in this study were 7.59 ± 3.55% (range: 1.83–18.40%), 0.89 ± 0.86% (range: 0.04–4.73%), and 0.67 ± 0.53% (range: 0.09–3.30%), respectively. M-MDSC was significantly correlated with WBC (r = 0.239, *p* = 0.016) and CRP level (ρ = 0.266, *p* = 0.007), again with no correlation with PSA level (Appendix A). The percentage of M-MDSC in the cancer group was higher than in the control group (*p* = 0.056), but there was no difference in total MDSC or PMN-MDSC. There was no significant difference observed in M-MDSC among different cancer grades (Appendix A). M-MDSC was initially divided into three groups, following the same methodology as LTL. However, given the non-parametric distribution of M-MDSC, it was eventually categorized into two groups: a low group (lower 2/3: <0.70%, n = 68) and high group (upper 1/3: ≥0.70%, n = 34). Multivariate logistic regression analysis showed that the high group had a significantly higher risk of prostate cancer diagnosis than the low group (OR = 3.031, 95% CI: 1.152–7.977, *p* = 0.025) (Table 3).

### 3.4. Leukocyte Telomere Length Correlates with the Percentage of M-MDSC

The association between LTL and M-MDSC was examined based on the above results. There was a significant correlation between LTL and MDSC (continuous or log-transformed), yielding correlation coefficients of ρ = −0.346 (*p* < 0.001) and r = −0.373 (*p* < 0.001), respectively. In partial correlation analysis, accounting for cancer status, the correlation coefficient between LTL and M-MDSC (log-transformed) was calculated as r = −0.347 (*p* < 0.001). Significance was also observed after adjusting for age, BMI, and smoking status. Moreover, LTL and M-MDSC (log-transformed) were divided into three quartiles each, resulting in nine segments. These segments were then compared, creating a group with short LTL and high MDSC (n = 15) against a group with long LTL and low MDSC (n = 17) (Figure 1). The two groups did not differ significantly in the prevalence of prostate cancer (*p* = 0.131). However, the former group had significantly elevated HbA1c levels (*p* = 0.009) and a higher prevalence of hypertension (indicated by using antihypertensive drugs, *p* = 0.014) and tended to have lower testosterone levels (*p* = 0.076) in comparison to the latter group.

## 4. Discussion

This study explored the relationship between LTL, MDSC, and prostate cancer. Our findings revealed that patients with localized prostate cancer exhibited significantly shorter LTL and a greater proportion of M-MDSC in comparison to the healthy control group, alongside variations in PSA levels and age. In addition, there has been no report on the correlation or simultaneous measurement of LTL and MDSC. We provided the first report (known to us) of a significant negative correlation between LTL and MDSC.

Recent research indicates that patients with shorter LTL in the pretreatment stage were approximately 2.7 times more likely to present with high-grade prostate cancer (Gleason score ≥ 8) than those with longer LTL (95% CI: 1.79–4.18; *p* = 3.11 × 10^−6^), and patients with a higher Gleason score had significantly shorter LTL. Additionally, these patients faced an increased risk of biochemical recurrence following total prostatectomy or radiotherapy (HR = 1.73, 95% CI: 1.08–2.78; *p* = 0.021) [16]. In a study on MDSCs and prostate cancer, cancer patients had an increased proportion of M-MDSCs (defined in this study as cells with CD14^+^ characteristics, as the gating of MDSCs differs in each article) compared to controls [19,30,31]. In patients with castration-resistant prostate cancer, those with higher MDSCs had shorter overall survival (OS) (19 months vs. 33 months; *p* < 0.05) [19]. Our results, along with prior studies, showed that shorter LTL and higher MDSC were associated with cancer risk, and patients with a higher Gleason score or a higher D’Amico risk classification had significantly shorter LTL compared to those with low-grade findings, with our additional finding that even localized prostate cancer has altered molecular biology and immunology in PBMC.

When cell division shortens telomeres to a critical length, it triggers the DNA damage response (DDR) mechanism. This response activates growth suppressor groups within the p53-p21 and p16-RB pathways. Consequently, this leads to the irreversible arrest of cell division and induces cellular senescence [32,33]. On the other hand, p21 and p16 are expressed in M-MDSCs, and they suppress the immune response to cancer by promoting the migration and accumulation of M-MDSCs into cancer tissue. In other words, p21 and p16, conventionally considered cancer suppressors, enhance cancer progression via MDSCs [34]. Although senescent cells do not proliferate, they remain metabolically active and significantly affect the tumor microenvironment at the secretome, a phenomenon known as senescence-associated secretory phenotype (SASP) [35,36]. SASP-associated secretions, including inflammatory cytokines and chemokines, stimulate myelopoiesis in the bone marrow and spleen. This stimulation increases MDSC production and hinders the elimination of cancerous and senescent cells. As a result, tissue homeostasis maintenance is disrupted, influencing tumorigenesis and accelerating the progression of age-related diseases [7,37,38]. TRF2 (telomere repeat binding factor 2), one of the factors that make up the telomere terminal protein complex shelterin, also plays an essential role in the protection and maintenance of telomeres, regulation of telomere length, and chromosome stability [39,40]. Dysfunction or aberrant expression of TRF2 contributes to tumorigenic effects by causing chromosome instability through telomere shortening [41] and induces cancer progression and metastasis by causing accumulation of MDSCs in cancer cells. Indeed, analyses of cancer cohorts (breast, stomach, ovarian, and lung cancers) showing a correlation between TRF2 expression and MDSC invasion have shown an inverse correlation with the overall survival of patients [42]. Furthermore, in our data, a history of hypertension, diabetes, and low serum testosterone levels was associated with short telomeres and high MDSC. Oxidative stress serves as a common underlying factor in various conditions. Hypertension, diabetes, and their complications as well as low testosterone levels are known to induce disease onset and progression through oxidative stress, leading to vascular aging and increasing the risk of lifestyle-related diseases and cancer [43,44,45]. Additionally, telomeres are susceptible to damage from oxidative stress, which leads to DNA damage. This accumulation of damage causes telomere shortening and dysfunction, impacting cellular health [46,47]. Oxidative stress induces MDSCs by activating NF-κB, a key transcription factor in immune responses, thereby increasing MDSC percentages [48]. Our findings align with the anticipated correlation between telomere shortening and the rise in MDSCs.

Additionally, older age not only heightens the risk of developing prostate cancer but also of its aggressive forms, necessitating early diagnosis and treatment [49]. Traditional prostate cancer detection methods based on elevated PSA levels often lead to overdiagnosis and overtreatment, raising cost-effectiveness concerns [50]. Measuring LTL and MDSC via minimally invasive blood tests could enhance the accuracy of patient selection for biopsies. Using these measurements as adjunct biomarkers could alleviate patient burdens and contribute to reduced medical expenses.

A key strength of this study is the timing of LTL and MDSC measurements, conducted just before diagnosis. Unlike previous LTL research on prostate cancer, which involved patients already diagnosed or long-term prospective studies with significant delays between specimen collection and diagnosis, our study’s measurements were contemporaneous with the clinical assessment [51,52,53]. Moreover, while past MDSC studies often included post-treatment patients [19,30], our research uniquely focused on pretreatment individuals, with specimens collected a day before prostate biopsy, ensuring timely and relevant data. In addition, limiting the survey to localized prostate cancer avoided significant differences in the background of the patient population, including disease status. The second feature was the accuracy of the LTL measurements. In a review of telomere length assays, the accuracy (reproducibility) of the PCR assay ranged from 2.5% to 12% for intra-assay CV, 3.97% to 15.9% for inter-assay CV, and 0.89 and 0.92 for ICC, which were all equal or better in this study [54]. In 2019, the Telomere Research Network (TRN) proposed Minimum Reporting Recommendations for PCR-based Telomere Length Measurement (https://trn.tulane.edu/) for the purpose of establishing a standard telomere measurement applicable to telomere researchers. The LTL measurements in this study largely met the TRN recommended criteria, including high quality of DNA samples, sample size, and ICC analysis.

This study has several limitations to acknowledge. Firstly, the small sample size of 102 blood samples and potential data bias, particularly in age and PSA levels between cancer and healthy patients, could affect the results. To mitigate this, age and PSA levels were used as confounding factors in our statistical analysis. Secondly, our study was confined to a Japanese cohort. Given the reported ancestral differences in LTL, our findings may not be generalizable across different ancestries [55]. Thirdly, we did not measure cytokines and oxidative stress markers, which could have provided more profound insights into the LTL-MDSC correlation. Future studies encompassing larger and more diverse cohorts along with additional measurements are warranted to address these limitations comprehensively and delve deeper into exploring the relationship between LTL and MDSC.

## 5. Conclusions

Our study is pioneering in revealing an association between LTL and M-MDSC in patients undergoing biopsy for prostate cancer, indicating that both factors may be potentially useful as adjunctive biomarkers for the early diagnosis of prostate cancer. Furthermore, we uncovered a direct negative correlation between LTL and M-MDSC, which might contribute to a deeper understanding of the molecular, biological, and immunological factors involved in cancer development. We intend to delve deeper into investigating causal relationships by genotyping the samples from this prospective observational study and employing Mendelian randomization, leveraging genetic information.

## Figures and Tables

**Figure 1 cancers-16-01386-f001:**
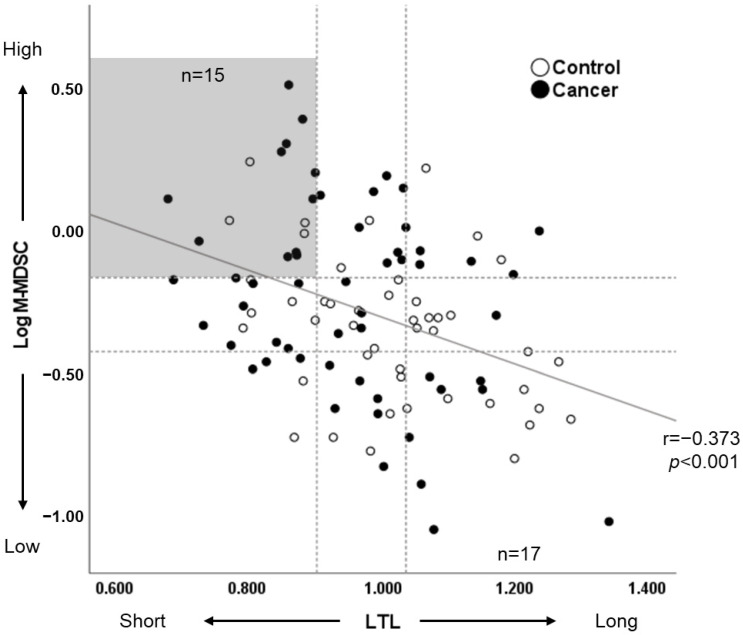
Correlation analysis between LTL and M-MDSC (logarithmic transformation of the data). Participants were divided into three groups based on tertile values. The group with short LTL and high MDSC (n = 15) and the group with long LTL and low MDSC (n = 17) are highlighted in gray.

**Table 1 cancers-16-01386-t001:** Baseline characteristics of patients with biopsy results.

Characteristics	Total (N = 102)		Cancer (N = 56)				Control (N = 46)				Mean Difference (*t*-Test/M-W)		
	Mean/*n* (%)	SD	Mean/*n* (%)	Min	Max	SD	Mean/*n* (%)	Min	Max	SD	t/Z	df	*p*-Value
Age (years)	67.9	8.4	70.0	56.0	87.0	7.9	65.3	47.0	81.0	8.4	−2.869	100	**0.005**
BMI (kg/m^2^)	24.2	2.8	24.1	18.5	31.7	3.0	24.3	18.7	30.0	2.5	0.346	100	0.730
Smoking status													
Smoking, *n* (%)	66 (64.7)	-	40 (71.4)	-	-	-	26 (56.5)	-	-	-	-	-	-
Non-smoking, *n* (%)	36 (35.3)	-	16 (28.6)	-	-	-	20 (43.5)	-	-	-	-	-	-
PSA (ng/mL)	10.6	13.3	13.1	3.4	116.4	17.3	7.5	2.3	22.2	3.7	2.206	-	**0.027 ***
Testosterone (ng/mL)	4.90	1.66	4.96	1.24	9.58	1.78	4.83	2.54	7.93	1.51	−0.398	100	0.692
CRP (mg/dL)	0.13	0.21	0.18	0.01	1.29	0.26	0.08	0.01	0.33	0.73	1.959	-	**0.050 ***
HbA1c (%)	5.9	0.6	5.9	5.2	8.2	0.6	5.8	5.3	7.5	0.5	1.342	-	0.180 *
LTL	0.980	0.142	0.951	0.675	1.342	0.141	1.015	0.768	1.265	0.138	2.290	100	**0.024**
Total MDSC (%/PBMC)	7.59	3.55	7.29	2.69	18.40	3.57	7.95	1.83	15.60	3.53	−1.288	-	0.198 *
PMN-MDSC (%/PBMC)	0.89	0.86	0.80	0.10	4.73	0.89	1.01	0.04	3.23	0.83	−1.641	-	0.101 *
M-MDSC (%/PBMC)	0.67	0.53	0.78	0.09	3.30	0.62	0.54	0.16	1.77	0.36	1.910	-	0.056 *

* Due to their non-normal distributions, the assessment was conducted using the Mann–Whitney U-test.

**Table 2 cancers-16-01386-t002:** Short LTL associated with high risk of prostate cancer.

LTL *	β	Adjusted OR **	95% CI	*p*-Value
Short	1.092	2.979	1.001, 8.869	**0.050**
Middle	0.660	1.934	0.661, 5.658	0.228
Long	Reference			

* LTL was categorized into three groups based on continuous values: short (<0.900, n = 34), middle (0.900–1.035, n = 34), and long (>1.035, n = 34). ** Adjusted by age, BMI, smoking status, and PSA levels.

**Table 3 cancers-16-01386-t003:** High M-MDSC level associated with high risk of prostate cancer.

M-MDSC *	β	OR **	95% CI	*p*-Value
High	1.109	3.031	1.152, 7.977	**0.025**
Low	Reference			

* M-MDSC was categorized into two groups based on continuous values: high (≥0.70%, n = 34) and low (<0.70%, n = 68). ** Adjusted by age, BMI, smoking status, and PSA levels.

## Data Availability

All data analyzed in this study can be provided by applying to the corresponding author, S. Horie.

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
