# Peer review of "Evaluating Leukocyte Telomere Length and Myeloid-Derived Suppressor Cells as Biomarkers for Prostate Cancer"

_cancers, 2024, doi:10.3390/cancers16071386_

Round 1
Reviewer 1 Report
Comments and Suggestions for Authors
The manuscript entitled “Evaluating Leukocyte Telomere Length and Myeloid-Derived Suppressor Cells as Biomarkers for Prostate Cancer” aimed to investigate the potential of leukocyte telomere length and myeloid-derived as diagnostic biomarkers for prostate cancer using patients undergoing a prostate biopsy. Based on the obtained results, the authors concluded that their findings could potentially contribute to a deeper understanding of the molecular, biological, and immunological factors involved in cancer development and that results suggested that LTL and M-MDSC could serve as additional biomarkers for the early diagnosis of prostate cancer.
In the Introduction section, when talking about the risk of cancer it could be worth mentioning also the increased cancer costs and other forgotten public health impacts of cancer especially in the aging population. Please see:
Viegas et al. Forgotten public health impacts of cancer - an overview. Arh Hig Rada Toksikol. 2017; 68(4): 287-297. doi: 10.1515/aiht-2017-68-3005.
In the Materials and Methods section please indicate ethical approval for the study (as stated in the Institutional Review Board Statement).
Could the results be a bit compromised by the fact that the cancer group exhibited a significantly higher age compared to the control subjects?
Besides, telomeres could be affected by different anthropometric, lifestyle, or genetic factors of different ethnic groups, therefore, this should be kept in mind concerning that the study was done on Japanese patients.
The authors state that there was no significant difference among LTL in the cancer group regarding clinical stage, Gleason score, and D'Amico risk classification but data are not shown. I would suggest providing data at least in the supplementary material. The same goes with other not shown data. This could be important in line with proposing new diagnostic/clinical biomarkers.
Minor remarks:
References should be placed before the full stops.
Please mind the spacing between numbers and units throughout the paper.
Use min instead of minutes.
Tables are inserted as Figures in the text.
Reviewer 2 Report
Comments and Suggestions for Authors
The manuscript investigates the association between leukocyte telomere length (LTL), monocytic myeloid-derived suppressor cells (M-MDSC), and prostate cancer. The study shows that patients with localized prostate cancer have shorter LTL and higher percentages of M-MDSC compared to healthy controls. They discover a significant negative correlation between LTL and M-MDSC levels. Additionally, shorter LTL and higher M-MDSC levels are associated with increased prostate cancer risk. The study concludes that LTL and M-MDSC could serve as biomarkers for early prostate cancer diagnosis. Overall, the manuscript provides valuable insights into the interplay between telomere length, MDSCs, and prostate cancer. To understand the complex interplay between LTL, M-MDSC, and prostate cancer, the authors need to provide a more comprehensive and functional analysis:
1) As described in the manuscript, there may be potential bias in the data, particularly regarding age and PSA levels between cancer and healthy patients. It is not surprise to see shorter LTL in aged group. It remains unclear whether, under the same age group, cancer patients exhibit shorter LTL and higher percentages of M-MDSCs. the authors could perform subgroup analyses stratified by age to determine if differences in LTL and M-MDSC percentages persist across age-matched cohorts of cancer patients and healthy controls. This approach would help elucidate whether observed associations are independent of age-related factors.
2) The correlation between LTL and M-MDSC does not provide valuable insights into underlying mechanisms. Although the authors mentioned the causation in the discussion, it would be more meaningful to explore molecular pathways associated with LTL and M-MDSC through gene expression dataset, including transcriptomic data from peripheral blood samples
3) Have the authors perform the follow-up studies? It would be informative to investigate correlations between survival outcomes, treatment responses, and LTL/M-MDSC profiles in cancer patients. The results could elucidate whether LTL and M-MDSC levels predict disease progression, treatment efficacy, or overall survival.
Round 2
Reviewer 1 Report
Comments and Suggestions for Authors
Authors answered all the raised comments and questions and changed their paper substantially.